# Sustained induction of autophagy enhances survival during prolonged starvation in newt cells

Md Mahmudul Hasan[1], Shinji Goto[1], Reiko Sekiya[1], Toshinori Hayashi[2,3], Tao-Sheng Li[1], Tsuyoshi Kawabata[1]

**Salamanders demonstrate exceptional resistance to starvation, allowing them to endure extended periods without food in their natural habitats. Although autophagy, a process involving evolutionarily conserved proteins, promotes survival during food scarcity, the specific mechanism by which it contributes to the extreme starvation resistance in newt cells remains unexplored. Our study, using the newt species *Pleurodeles waltl*, reveals that newt primary fibroblasts maintain constant autophagy activation during prolonged cellular starvation. Unlike normal mammalian fibroblasts, where autophagosome formation increases during acute starvation but returns to baseline levels after extended periods, newt cells maintain elevated autophagosome numbers even 4 d after autophagy initiation, surpassing levels observed in nutrient-rich conditions. Unique *P. waltl* mTOR orthologs show reduced lysosomal localization compared with mammalian cells in both nutrient-rich and starved states. However, newt cells exhibit dephosphorylation of mTOR substrates under starvation conditions, similar to mammalian cells. These observations suggest that newts may have evolved a distinctive system to balance seemingly conflicting factors: high regenerative capacity and autophagy-mediated survival during starvation.**

## Introduction

Resistance to starvation is a prerequisite for the survival of animals. Across vertebrates, various species have adapted to starvation conditions, providing strategies to suppress the detrimental effects (McCue, 2010). Although common vertebrates are often forced to survive without food, some subterranean animals are exposed to extremely long periods of starvation because of limited access to food supplies. For example, the water frog *Rana esculenta* survives 12 mo of starvation (Grably & Piery, 1981), and the African clawed toad *Xenopus laevis* survives after 18 mo of starvation (Merkle & Hanke, 1988) because these animals are often exposed to extremely reduced food supplies. The cave-dwelling salamander *Proteus anguinus* has been known to survive food deprivation for 18–96 mo (Hervant et al, 2001). Ectothermic amphibians are also highly resistant to starvation. The Iberian ribbed newt *Pleurodeles waltl* (*P. waltl*) survives starvation for over 19 mo (Peng et al, 2021). Because autophagy, an intracellular degradation system, is known to play an essential role in survival during starvation by supplying minimum backup energy and building blocks to maintain cellular and tissue homeostasis in a variety of organisms (Mizushima & Levine, 2010; Guo & White, 2017; Kawabata & Yoshimori, 2020), it is postulated that this is also the case in newts. In fact, autophagy-related genes are highly expressed during starvation in newts, and inhibiting autophagy with a lysosomal inhibitor reduces newt cell proliferation during tissue regeneration (Peng et al, 2021). These data suggest that autophagy may be required for the newt survival during starvation. As autophagy is known to suppress a wide range of diseases, including neurodegeneration, it is an attractive idea that the characteristic autophagic activity of newts could be beneficial to human health. However, the molecular mechanisms underlying newt autophagy remain largely unknown. In mammalian cells, an in vitro study showed that nutrient starvation strongly induces autophagy for a few hours, followed by attenuation of autophagosome formation (Yu et al, 2010; Rong et al, 2011). This attenuation is required for lysosome reformation (Yu et al, 2010; Yim & Mizushima, 2020). However, its contribution to survival during long-term starvation in vivo remains unclear.

In contrast to the limited regenerative capacity of mammals, many non-vertebrate organisms have a remarkable capacity for tissue regeneration (Mehta & Singh, 2019). Regeneration of lost tissue in newts involves several steps, including proliferation of stem cells, dedifferentiation of cells in the region adjacent to the lost tissue, and their differentiation (Hayashi et al, 2013; Tanaka et al, 2016; Mehta & Singh, 2019). Newt regeneration requires several pathways involved in development and adult tissue maintenance, such as the FGF receptor, Hedgehog, and Wnt pathways (Del Rio-Tsonis et al, 1998; Singh et al, 2012), in which autophagy is tightly linked. In addition, because autophagy is involved in the regeneration of dissected caudal fins of zebrafish, promoting differentiation and suppressing apoptosis (Varga et al, 2014), it is necessary to determine how autophagy is regulated in newt cells; however,

[1]Department of Stem Cell Biology, Atomic Bomb Disease Institute, Nagasaki University, Nagasaki, Japan [2]Graduate School of Integrated Sciences for Life, Hiroshima University, Hiroshima, Japan [3]Amphibian Research Center, Hiroshima University, Hiroshima, Japan

Correspondence: t-kawabata@nagasaki-u.ac.jp

 

the molecular mechanism involved in this regulation has not been fully elucidated (Petherick et al, 2013; Cinque et al, 2015; Lorzadeh et al, 2021).

Here, we show that newt cells have a unique autophagy regulatory system that allows them to maintain induced levels of autophagosome biogenesis for a prolonged period, significantly longer than in normal mammalian cells. These findings may explain the differences in tolerance of mammals and newts to long-term starvation.

# Results

### Primary newt fibroblasts retain a greater rate of autophagosome biogenesis under prolonged starvation than normal mammalian fibroblasts

To understand the molecular mechanisms underlying how newts survive for more than a year without feeding, we sought to determine the kinetics of starvation-induced autophagosome formation in newt cells. We used the newt strain *P. waltl*, which has been established as a model newt for genetic analysis. *P. waltl* is highly reproductive, easy to maintain, has a relatively short generation time, and produces many eggs throughout the year (Hayashi et al, 2013). We established a suitable method to generate and maintain primary newt fibroblasts from newt limbs (Hasan et al, 2023). Newt cells were starved for 2–24 h, harvested, and subjected to Western blotting or immunocytochemistry using an antibody directed against LC3, the most common marker for autophagosomes. Induction of autophagic flux, as measured by conventional LC3 Western blot using lysosomal inhibitor bafilomycin A1 (BafA1), showed that mammalian cells exhibited a plateau level of autophagy around 2 h after starvation, followed by a downward trend in autophagy levels at 6–24 h (Fig S1A). Newt cells also showed an increase in autophagic flux at 2 h after induction, followed by a more increase even at 6 h after induction and reduction at 24-h timepoint (Fig S1B). This suggests that both mammalian and newt cells are able to rapidly induce autophagy in response to nutrient starvation, but newt cells may show slightly slower and gradual induction kinetics of autophagy. Note that the expression levels of endogenous LC3 decreased significantly 6–24 h after starvation induction because of degradation of LC3 by autophagy itself, resulting in an underestimate of autophagic flux in the prolonged starvation condition. To determine the difference in autophagic activity in newt cells and mammalian cells during prolonged starvation, we counted the number of autophagosomes, as indicated by the formation of LC3 puncta. The number of LC3 puncta increased significantly 2 h after starvation, followed by a gradual increase at 4 and 6 h after induction in newt cells, a peak at 6-h timepoint, and a slight decrease at 24 h (Fig 1A and B). It is noteworthy that newt cells retained the induced level of the number of autophagosomes even at 96 h of starvation. Normal human dermal fibroblasts (HDF) showed quick and significant induction of autophagosome formation and reached a plateau at 2–6 h post-induction, followed by a return to a level comparable to the nutrient-rich condition at 24 h, which continued until 96 h post-

induction (Fig 1C and D). Pan-cytoplasmic background distribution of LC3 remains lower than the nutrient-rich condition until the 96-h timepoint. It indicates that HDF show quick induction of autophagy at 2 h after starvation and become close to the steady-state level of autophagy at 24 h after starvation, which shows slightly the induced level of autophagy but not at the statistically significant level. Note that the size of newt fibroblasts is significantly larger than that of mammalian cells used in this study (Fig S2A); therefore, we show the number of autophagosomes per area (# of autophagosome/$\mu$m2). Two other human normal fibroblast cell lines, MRC5 and HE-1, showed a similar tendency to induction of LC3 dot formation at 2 h after starvation and become the steady-state level at 24 h after starvation (Fig S2B and C). Thus, newt cells retain an induced level of autophagic capacity during prolonged starvation, which is much longer than that of normal human fibroblasts.

Interestingly, the pancreatic cancer cell line PANC1 showed induction of autophagosome formation at a slightly slower pace than HDF (Fig 1C and E). PANC1 showed a significant increase in the number of autophagosomes and reached a plateau at 4 h post-starvation, and maintained higher autophagosome levels until 24 h post-starvation (Fig 1C and E). This is also similar to other cancer cell lines, A549 (adenocarcinoma, lung cancer) and NCI-H1299 (non–small-cell lung cancer), which showed substantially higher autophagosome numbers even at 24 h post-starvation (Fig S2B and C). This finding is consistent with the fact that cancer cells often rely on autophagy to survive in a harsh cancer microenvironment and exhibit increased autophagic activity (White, 2015). So, newt cells are somehow closer to mammalian cancer cell lines than normal mammalian cells in terms of the kinetics of autophagosome formation in response to nutrient starvation. We speculate that newt cells have evolved to adapt to crude circumstances that require prolonged autophagic activity during starvation, which is similar to what occurs in the cancer microenvironment at the molecular level.

As both the number and size of autophagosomes determine the activity of autophagy, we estimated the size of autophagosomes in newt cells. Similar to mammalian cells, whose autophagosomes are typically 0.5–2 $\mu$m in diameter (Klionsky & Eskelinen, 2014), newt fibroblasts are on average 0.9 $\mu$m under nutrient-rich conditions, but 1.4 $\mu$m 4 h after starvation induction (Figs S3 and S4A). Similar observations were made in mammalian cells, with HDF and PANC1 exhibiting enlarged autophagosomes during starvation compared with nutrient-rich conditions (Fig S4A). Newt cells displayed larger autophagosomes than HDF and PANC1 cells not only in nutrient-rich conditions but also at 4, 24, and 96 h after starvation (Figs S3 and S4B). This suggests that each autophagosome in newt cells may have a greater capacity to engulf its cytoplasmic components than in mammalian cells.

Notably, newt cells thrive at ~25°C, which is considerably lower than the optimal temperature for mammalian cells. Given that lower temperatures could potentially alter autophagosome biogenesis kinetics, we investigated the effects of culturing mammalian cells at reduced temperatures and newt cells at elevated temperatures. Because newt cells exhibit severe toxicity at higher temperatures, we examined autophagosome formation in mammalian cells at 25°C. We found that lower temperatures led to a marked increase in the number of LC3 dots, even under nutrient-rich conditions in mammalian cells, with no notable induction

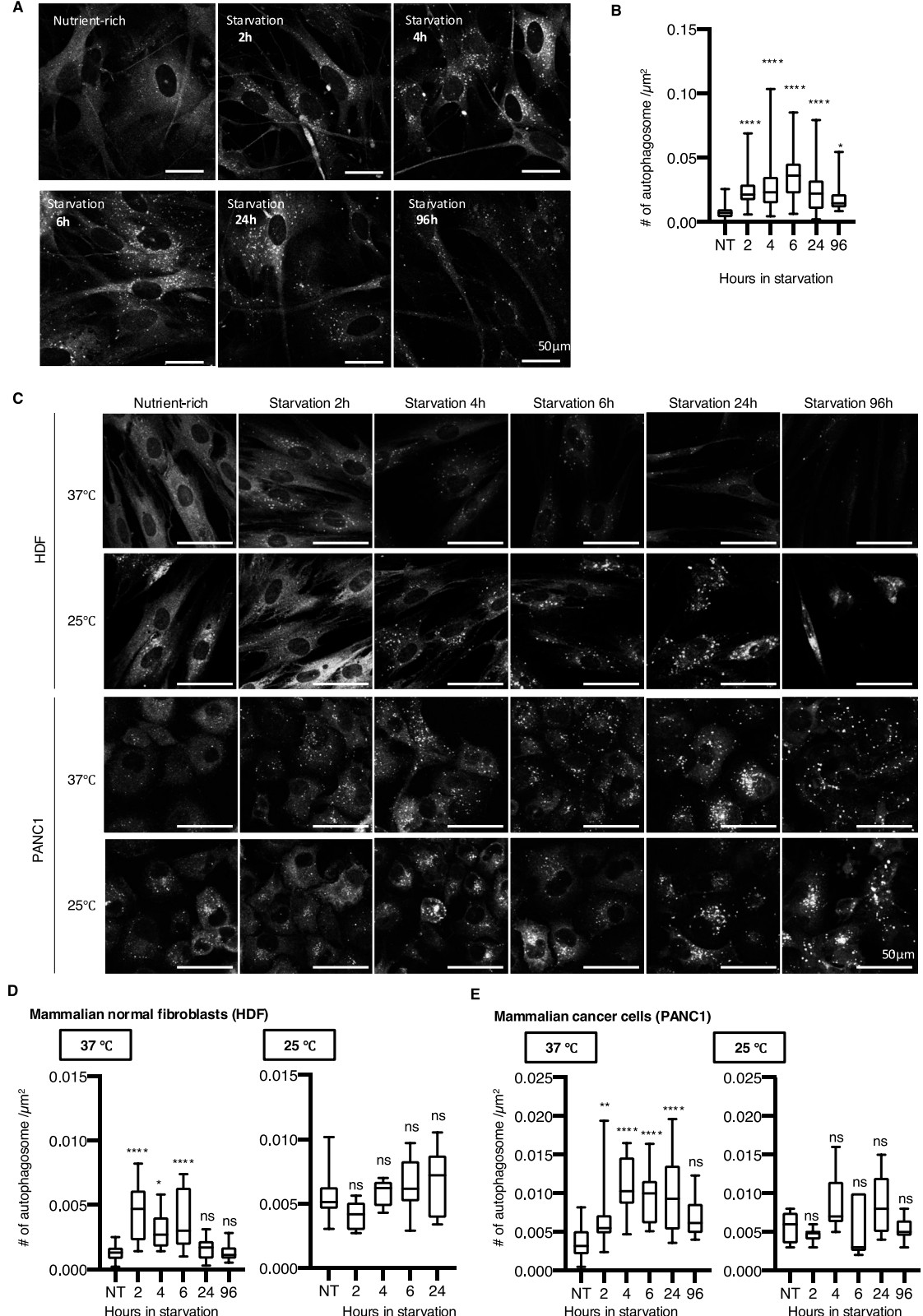

**Figure 1. Primary newt fibroblasts retain a greater rate of autophagosome biogenesis under prolonged starvation than normal mammalian fibroblasts.**
(A) Representative images of newt cells subjected to starvation for the indicated periods and stained with LC3 antibody. (B) Summarized boxplots showing the average number of autophagosomes per area ($/\mu m^2$) in newt cells. *$P < 0.05$, ****$P < 0.0001$ (one-way ANOVA, Dunnett's multiple comparisons test). (C) Representative images of HDF and PANC1 cells subjected to starvation for the indicated periods and stained with LC3 antibody. (D) Summarized boxplots showing the average number of

observed during starvation throughout the test period (Fig 1D and E). This is likely attributable to a cold stress response in which autophagy might enhance cellular survival under harsh conditions. Nevertheless, we can at least state that mammalian cells did not display increased autophagic activity 96 h post-starvation, even at lower temperatures, which stands in stark contrast to newt cells.

To rule out the possibility that the increase in the number of autophagosomes in newt cells exposed to prolonged starvation was simply due to an accumulation of autophagosomes caused by inhibition of lysosomal activity (inhibition of fusion between autophagosomes and lysosomes may lead to the accumulation of autophagosomes), we determined whether the formation of LC3 puncta 24 h after starvation in newt cells could be halted at a specific stage of autophagosome maturation. We evaluated the colocalization of LC3 with the lysosome marker LysoTracker and found that the colocalization rate 24 h after starvation was not significantly different from that at 2 or 4 h after starvation induction (Fig 2A–C), consistent with the similar observation as HDF and PANC1 cells (Fig S5A and B). It suggests that autophagosome maturation normally occurs 24 h after starvation in newt cells. Furthermore, we sought to test whether newt cells retain proper autophagosome maturation rate even in the prolonged starvation condition. Because of the difficulty in the use of the conventional assay such as tandem-fluorescent LC3 (Kimura et al, 2007) or autophagic flux probe using GFP-LC3-RFP-LC3ΔG (Kaizuka et al, 2016) as the mammalian system, we used reagents DAP red (autophagosome) and DAL green (autolysosome), which have been used to monitor autophagic flux in zebrafish (A562; Dojindo) (Sakurai et al, 2023). We found newt cells showed an increase in DAP red signal (autophagosome and autolysosome) at 4, 8, and 24 h after starvation, consistent with the increase in the number of LC3 punctum formation in the starved condition (Fig S6). DAL green (autolysosome) signal was gradually increased in a time-dependent manner in the starved condition, suggesting that autophagic flux is not halted even during prolonged starvation.

### The activity of the characteristic *P. waltl* mTOR orthologs is inhibited during prolonged starvation to maintain autophagy

To elucidate the molecular mechanism underlying autophagy induction in newt cells, we investigated the activity of the primary autophagy regulator. mTOR and its counterparts are recognized as crucial autophagy modulators across various organisms, functioning through phosphorylation-mediated inactivation of autophagy regulators like ULK1 and AMPK (Liu & Sabatini, 2020). Recent findings have revealed that axolotls possess a specialized regulatory system involving axolotl mTOR. This system contains urodele-specific insertions that render endogenous axolotl mTOR intracellular localization on the lysosome and priming of this pathway for rapid activation (Zhulyn et al, 2023). Notably, introducing axolotl-like mTOR into mammalian cells leads to persistent mTOR localization on lysosomes during nutrient deprivation,

potentially facilitating tissue regeneration even under starvation conditions. We obtained the mRNA sequence of *P. waltl* mTOR from the iNewt database (http://www.nibb.ac.jp/imori/main/) (Matsunami et al, 2019) and created alignments of mTOR orthologs from human, axolotl, and *P. waltl*. Our analysis revealed that the mTOR in the newt cells used in this study (referred to as PlemTOR) contains identical insertion amino acid sequences as axolotl mTOR (insertion#1: SHQPSPQ, and insertion#2: ELKTDVLETTDPLRTDSNK) (Fig 3A). Intriguingly, *P. waltl* possesses an additional mTOR variant that includes only insertion#2 but lacks insertion#1, suggesting that *P. waltl* may employ an axolotl-like yet distinct regulatory system using different PlemTOR variants (Fig 3A).

Given that mTOR is active on lysosomes, we investigated how mTOR localizes to lysosomes during starvation and discovered that PlemTOR localization on lysosomes in newt cells differs from that in mammalian cells. In nutrient-rich conditions, mTOR predominantly localizes to lysosomes in mammalian HDF and PANC1 cells (Fig 3B and C). This lysosomal localization decreases during starvation, even after 24 h. The size of lysosomes was only slightly changed during starvation in mammalian cells (Fig S7B). Conversely, newt cells exhibit only partial PlemTOR localization on lysosomes, even in nutrient-rich conditions (Fig 3B and C). Moreover, PlemTOR lysosomal localization did not change significantly during starvation for up to 24 h. This could be partly attributed to an increase in lysosome number, but not size, during starvation in newt cells (Figs 2D and S7A and B). Consequently, PlemTOR's lysosomal localization may be underestimated in nutrient-rich conditions. Nevertheless, the lack of changes in PlemTOR's lysosomal localization during starvation might indicate that mTOR is starvation-resistant, as suggested by a previous study showing that changes in lysosomal localization of axolotl-mimic mTOR are less sensitive than mammalian mTOR when expressed in mammalian cells (Zhulyn et al, 2023). To determine how PlemTOR activity changes in newt cells during starvation, we assessed the phosphorylation status of mTOR substrates in these cells. The mTOR-mediated phosphorylation of ULK1 at Ser757, which is essential for autophagy suppression during nutrient abundance and is reduced during starvation to rapidly initiate autophagosome formation, showed a significant decrease in newt cells as early as 2 h after starvation. This was followed by only a minor increase from 4 to 96 h post-starvation (Fig 3D). This response closely resembles that observed in two other mammalian cell types, HDF and PANC1 cells (Fig 3D), indicating that newt mTOR-dependent autophagy inhibition is swiftly terminated in response to starvation signals. In line with this, the phosphorylation of S6 kinase at Shr389, a general transcriptional regulator necessary for protein synthesis and a well-known mTOR substrate, also exhibits significant dephosphorylation as early as 2 h after starvation (Fig 3D). These findings lead us to conclude that newts possess a unique regulation of mTOR that enables hyper-regenerative capacity even under starvation conditions, while still maintaining appropriate autophagic activity to survive during nutrient deprivation. We hypothesize that the distinct intracellular localization of mTOR in

---

autophagosomes per area ($/\mu m^2$) in HDF and PANC1 in the indicated temperatures. ns, $P > 0.05$, $*P < 0.05$, $**P < 0.01$, $****P < 0.0001$ (one-way ANOVA, Dunnett's multiple comparisons test).

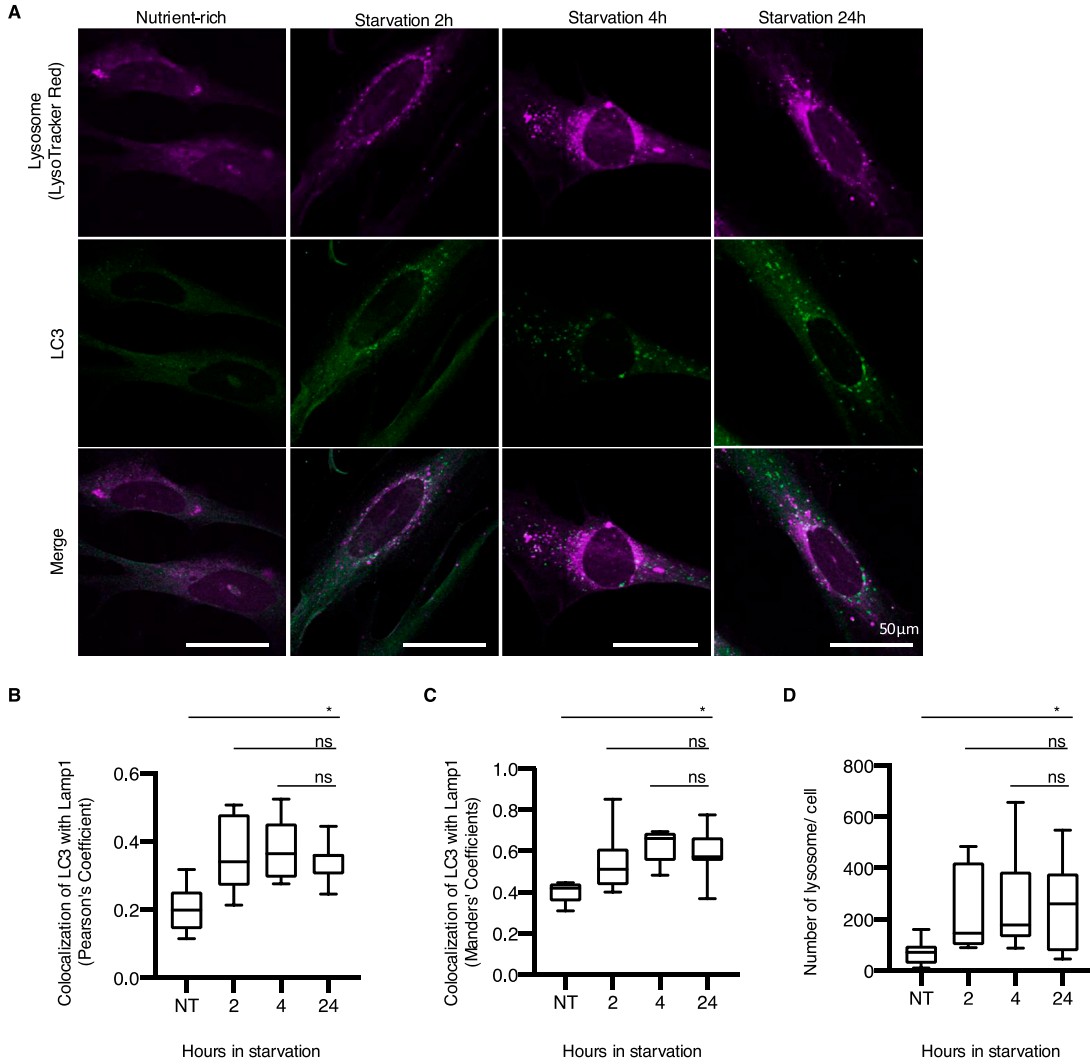

**Figure 2.   Colocalization of LC3 with lysosome in newt primary fibroblasts.**
**(A)** Representative images of newt cells stained with LC3 antibody and LysoTracker Red. Scale bar: 50 μm. **(B)** Summarized box plots showing the colocalization of LC3 with lysosome in newt cells indicated by Pearson's coefficient. ns, $P > 0.05$, *$P < 0.05$ (one-way ANOVA, Dunnett's multiple comparisons test). **(C)** Summarized box plots showing the colocalization of LC3 with lysosome in newt cells indicated by Manders' coefficient. ns, $P > 0.05$, *$P < 0.05$ (one-way ANOVA, Dunnett's multiple comparisons test). **(D)** Summarized box plots showing the number of lysosomes in newt cells. ns, $P > 0.05$, *$P < 0.05$ (one-way ANOVA, Dunnett's multiple comparisons test).

newts might contribute to managing both regenerative capacity and autophagy during starvation, although the exact mechanism requires further investigation in future studies.

Given the higher autophagic activity in newts during extended periods of starvation compared with mammalian cells, we hypothesized that inhibiting autophagy might severely disrupt newt cells under starved conditions. To test this, we evaluated cell survival during starvation with and without an autophagy inhibitor. Without the inhibitor, we did not observe any clear indication of newt cells being exceptionally resistant to starvation, as both HDF and PANC1 cells showed no significant decrease in cell numbers during the first 96 h of starvation (Fig 4A). However, after 1 wk, although HDF and PANC1 cells experienced substantial reductions (55% and 76%, respectively) from the 96-h timepoint, newt cells only showed a 36% reduction. This suggests that newt cells maintain a higher capacity to

endure prolonged starvation, aligning with the known starvation resistance of newts (Fig 4A). To investigate the role of autophagy in starvation survival, we cultured cells with or without BafA1. Surprisingly, newt cells did not exhibit increased sensitivity to BafA1 during starvation compared with HDF and PANC1 cells; instead, they demonstrated a slight resistance to the inhibitor (Fig 4B and C). This indicates that despite their elevated autophagic activity during extended starvation, newt cells may possess significant alternative mechanisms to survive starvation without relying on autophagy.

## Discussion

Newts, faced with the challenge of surviving in harsh conditions, have developed adaptations to cope with extreme starvation.

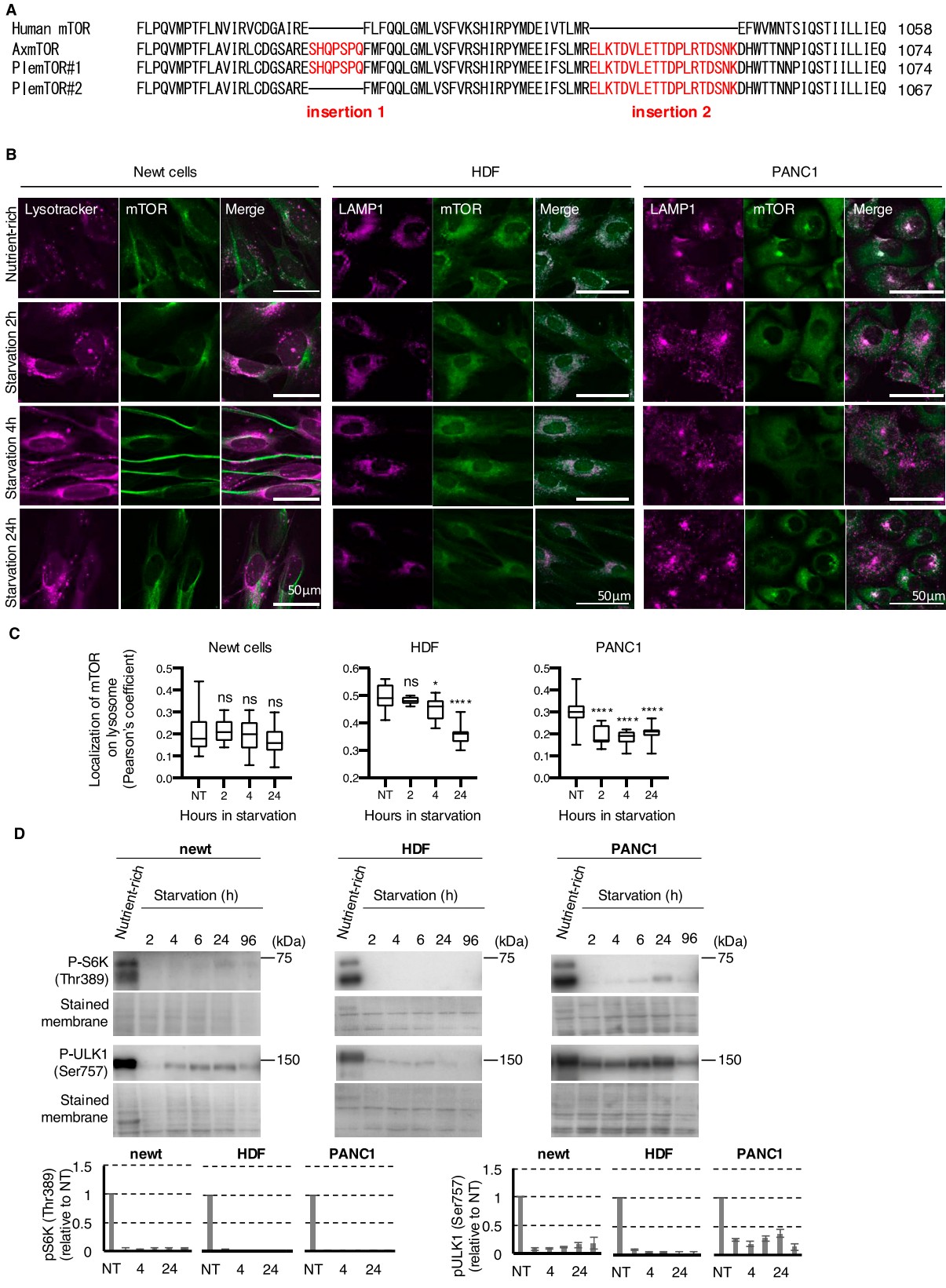

**A**

| | | |
|---|---|---|
| Human mTOR | FLPQVMPTFLNVIRVCDGAIRE————FLFQQLGMLVSFVKSHIRPYMDEIVTLMR————————EFWVMNTSIQSTIILLIEQ | 1058 |
| AxmTOR | FLPQVMPTFLAVIRLCDGSARESHQPSPQFMFQQLGMLVSFVRSHIRPYMEEIFSLMRELKTDVLETTDPLRTDSNKDHWTTNNPIQSTIILLIEQ | 1074 |
| PlemTOR#1 | FLPQVMPTFLAVIRLCDGSARESHQPSPQFMFQQLGMLVSFVRSHIRPYMEEIFSLMRELKTDVLETTDPLRTDSNKDHWTTNNPIQSTIILLIEQ | 1074 |
| PlemTOR#2 | FLPQVMPTFLAVIRLCDGSARE————FMFQQLGMLVSFVRSHIRPYMEEIFSLMRELKTDVLETTDPLRTDSNKDHWTTNNPIQSTIILLIEQ | 1067 |

insertion 1 · insertion 2

**B**

Newt cells · HDF · PANC1

**C**

Newt cells · HDF · PANC1

Localization of mTOR on lysosome (Pearson's coefficient)

Hours in starvation

**D**

newt · HDF · PANC1

Nutrient-rich · Starvation (h)

2 4 6 24 96 (kDa)

P-S6K (Thr389) —75

Stained membrane

P-ULK1 (Ser757) —150

Stained membrane

pS6K (Thr389) (relative to NT)

newt · HDF · PANC1

NT 4 24

Starvation (h)

pULK1 (Ser757) (relative to NT)

newt · HDF · PANC1

NT 4 24

Starvation (h)

Although humans maintain blood glucose levels around 5 mM under normal circumstances, with levels below half of this becoming problematic and one-third potentially life-threatening, newts exhibit a different pattern. Although their normal glucose levels are similar to humans, newts can sustain themselves at just 37% of their usual glucose levels during extended periods without food (Peng et al, 2021). This ability to function with significantly reduced glucose levels suggests that newts have evolved mechanisms to maintain homeostasis despite limited nutrient and energy availability. In this study, we demonstrated that newt cells produce autophagosomes at a higher rate than mammalian cells when subjected to prolonged starvation conditions. The differences in this matter due to the different culture conditions between newt cells and mammalian cells should be carefully discussed based on the different circumstances that exist in nature. Some differences in nutrient factors (e.g., 5 μg/ml insulin in newt medium) and osmotic differences should be taken into account. The appropriate temperature for culturing newt cells is about 20–26°C, not higher than 28°C, which is far from the 37°C required for culturing mammalian cells (Hayashi & Takeuchi, 2015; Hasan et al, 2023). Nevertheless, we observed that newt fibroblasts were able to rapidly induce autophagy, which is comparable to that of mammalian cells. The rate of autophagosome biogenesis in newt cells at 25°C was maintained at almost the same level from 2 to 24 h, followed by a slightly reduced but significantly higher rate than the nutrient-rich condition, suggesting that the higher autophagic activity of newt cells during prolonged starvation is not merely due to a delayed response, but also relies on its characteristic feature related to adaptation to prolonged starvation in natural situations.

The capacity of newt cells to maintain elevated autophagic activity during extended periods of starvation may seem at odds with their ability to regenerate tissue under limited food conditions in their natural habitat. This is because mTOR positively influences cellular proliferation and protein synthesis while negatively regulating autophagy. A previous study has shown that axolotl's mTOR's insensitivity to starvation contributes to its enhanced tissue regenerative ability, which is linked to increased proliferative activity at injury sites. Our findings demonstrate that newt primary fibroblasts can sustain higher autophagic activity during prolonged starvation, correlating with decreased mTOR activity under starved conditions. We hypothesize that this could be attributed to a unique characteristic of *P. waltl*, which possesses both an axolotl-type mTOR and an alternative variant containing only insertion#2 but lacking insertion#1 (PlemTOR#2) (Fig 3A). Because insertion 1 is located at the site necessary for RHEB-mediated mTOR activation (Long et al, 2007; Zhulyn et al, 2023), this alternative PlemTOR#2 might be responsible for triggering autophagy during extended starvation in newts. Further investigation is needed to understand how these two alternative mTOR variants work together to support

survival and tissue regeneration in newts during prolonged periods of food scarcity.

Despite the high levels of autophagic activity observed during prolonged starvation in newt cells, we found that inhibiting autophagy using the lysosomal inhibitor BafA1 led to a smaller reduction in cellular viability in newt fibroblasts compared with human normal fibroblasts. This suggests that autophagy is not the sole major factor determining the viability of newt cells during prolonged starvation, highlighting a key difference between newt cells and mammalian cells. It has been suggested that the cave-dwelling salamander *P. anguinus* exhibits extreme resistance to long-term food shortages because of its ability to reduce oxygen and energy consumption during starvation, along with possessing a high energy reserve (Hervant et al, 2001; Bizjak Mali et al, 2013). Observations of hepatocytes in this organism during fasting reveal morphological changes, including alterations in cell size and modifications in organelles such as lipid droplets, mitochondria, and autophagic vacuoles (Bizjak Mali et al, 2013). A comparison of two distinct populations of *Calotriton asper* (the Pyrenean mountain newt), one residing in subterranean habitats and the other in epigean environments, shows that cave colonization reduces metabolism while enhancing the accumulation of energy reserves. This adaptation allows for better survival during unpredictable fasting periods (Issartel et al, 2010). Because of the ability of newts to adjust their energy consumption and storage during extremely harsh conditions, we believe that although autophagy plays a role in survival during starvation, newts can still endure starvation through a coordinated regulation of their metabolism, even in the absence of functional autophagic activity.

In summary, our findings align with this survival mechanism employed by newts when faced with food scarcity. We hypothesize that the continuous suppression of mTOR during periods of starvation may be a tactic used by newts to persistently maintain elevated autophagy levels, enabling them to endure extended periods without food. We suggest that artificially modifying mammalian cells to replicate the newt's mTOR signaling pathway for autophagy could potentially serve as an innovative method for breaking down harmful targets responsible for various human diseases.

# Materials and Methods

### Animals

Male *P. waltl* were used in this experiment. Animals were collected from the Tottori University and the Hiroshima University Amphibian Research Center through the National BioResource Project (NBRP) of the Ministry of Education, Culture, Sports, Science and Technology (MEXT), Japan. The study was approved by the Institutional

**Figure 3. Newt cells exhibit decreased mTOR activity during prolonged starvation.**
**(A)** Alignment of the part of amino acid sequences of mTOR orthologs of human, axolotl, and the *P. waltl* showing the presence of characteristic inspersions in axolotl and newts. **(B)** Representative images of the newt cells, HDF, and PANC1 cells exposed to starvation for the indicated periods and stained with LysoTracker Red or LAMP1 (purple) and an mTOR antibody (green). Scale bar: 50 μm. **(C)** Summarized box plots of the colocalization shown by Pearson's coefficient analysis using at least 10 images from three independent analyses. ns, $P > 0.05$, *$P < 0.05$, **$P < 0.01$, ****$P < 0.001$ (one-way ANOVA, Dunnett's multiple comparisons test). **(D)** Representative image (left) and quantification (right) of immunoblots showing the phosphorylation levels of the indicated proteins. The error bars indicate the SEMs of at least three independent experiments.

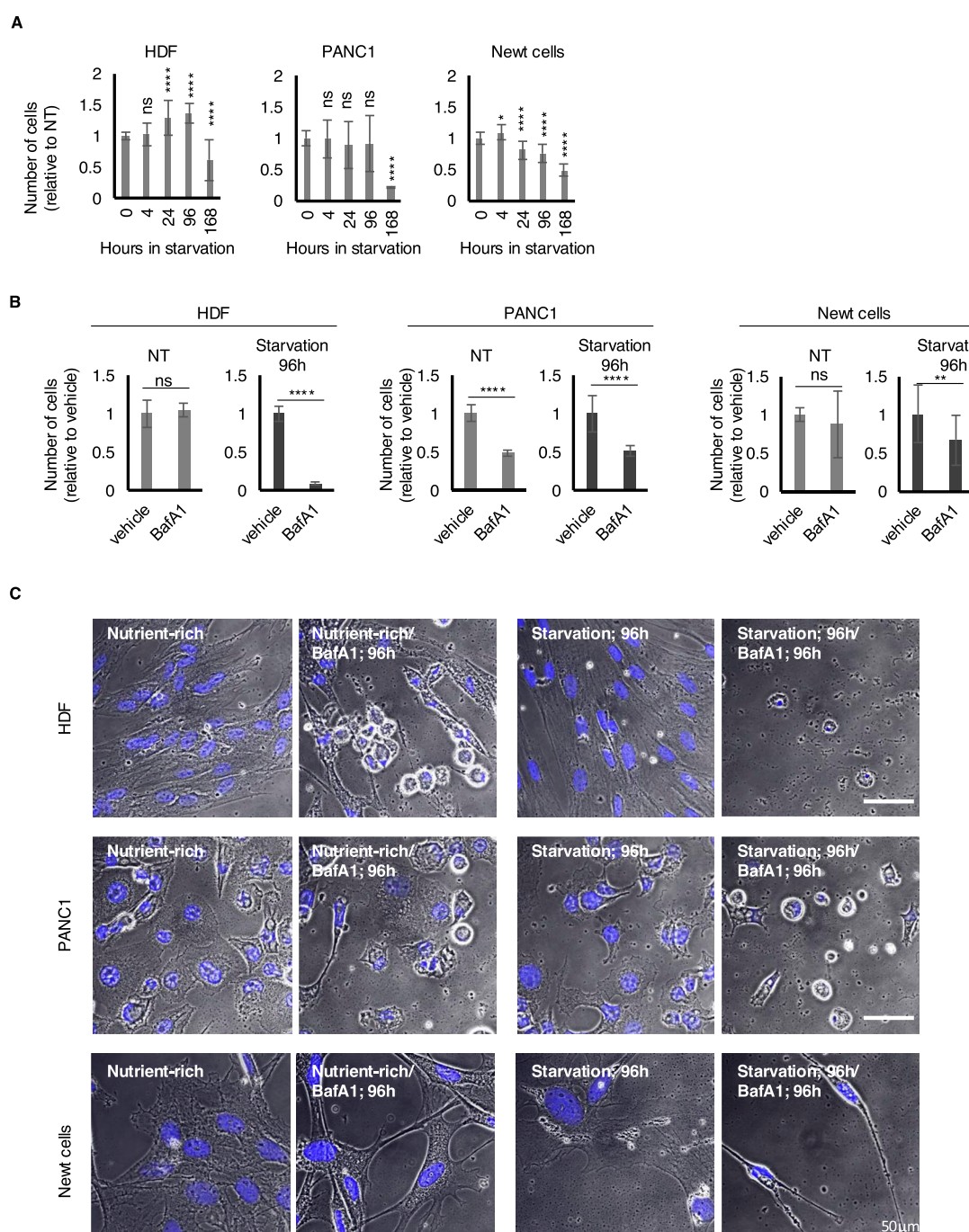

**Figure 4. Changes in the number of mammalian and newt cells after prolonged starvation.**
**(A)** Summarized bar graph showing the number of HDF, PANC1, and newt cells calculated by the number of nuclei in each observed area. At least nine independent pictures were analyzed for each experimental condition. Error bars show the SD. ns, $P > 0.05$, *$P < 0.05$, ****$P < 0.0001$ (one-way ANOVA, Dunnett's multiple comparisons test). **(B)** Summarized bar graph showing the number of cells treated with 125 nM of BafA1 for 96 h relative to vehicle-treated cells. At least nine independent pictures were analyzed for each experimental condition. Error bars show the SD. ns, $P > 0.05$, *$P < 0.05$, ****$P < 0.0001$ ($t$ test). **(C)** Representative DIC images (plus DAPI nucleus staining, blue) of cells exposed to 96 h of starvation with or without autophagy inhibitor BafA1. Scale bars: 50 μm. (Note that the scale of each picture of newt and mammalian cells is differently shown.)

Animal Care and Use Committee of Nagasaki University. All methods for animal experiments in this study were carried out in accordance with the guidelines (protocol code #2017-1) set forth by the Institutional Animal Care and Use Committee of Nagasaki University. All methods used are reported in accordance with the ARRIVE guidelines.

### Generation and culture of primary newt cells

Primary newt cells were grown and maintained according to methods developed in our laboratory (Hasan et al, 2023). Briefly, hind limbs were separated from the newt after the animal was anesthetized by submersion in 0.1% tricaine solution and washed with 10% povidone-iodine (Hasan et al, 2023). Thereafter, the limbs were washed thrice with Leibovitz's L-15 medium (catalog no. 128-06075; FUJIFILM Wako Pure Chemical Corporation) supplemented with 1% streptomycin–amphotericin B–penicillin. The limbs were then washed in a biological safety cabinet. The limbs were transferred to a culture dish filled with ice-cold L-15 medium and cut into small pieces of ~3 mm$^2$, after which the skin was removed from the tissue pieces. The tissues were subsequently washed six times, three times with L-15 medium and three times with RPMI 1640 (catalog no. 189-02025; FUJIFILM Wako Pure Chemical Corporation). A total of nine tissue pieces were seeded on a 6-cm culture dish containing 1.5 ml of RPMI 1640 medium. Importantly, the culture dish was coated with 2.5 $\mu g/cm^2$ fibronectin (FIBRP-RO; Sigma-Aldrich) before the tissues were seeded. The culture dish with tissues was transferred into an incubator set at 25°C and maintained at 2% $CO_2$ supply and humidity. Another 3.5 ml of medium was added to the culture dish overnight to allow attachment of the tissues to the culture surface. 2 ml of medium was added every 4 d. The cells were harvested two weeks after tissue seeding. For harvesting cells, the medium was discarded, and the culture surface was washed with 2 ml of amphibian phosphate-buffered saline twice before trypsinization by adding 1 ml of trypsin for 5 min. Thereafter, 2 ml of RPMI 1640 medium was added to inactivate the trypsin, and the cells were dissociated from the culture surface by gentle pipetting. The tissue pieces were discarded with forceps before centrifuging the cell suspension at 300$g$ for 3 min at room temperature. Then, the supernatant was discarded, and the cells were maintained on a fibronectin (2.5 $\mu g/cm^2$)-coated culture dish. Notably, the compositions of the L-15 medium, RPMI 1640 medium, amphibian phosphate-buffered saline, and working (1x) trypsin solution were modified in our laboratory to adjust the osmolality of the amphibians.

### Antibodies

We used antibodies against LC3 (PM036; MBL), mTOR (4517S; Cell Signaling Technology), LAMP1 (ab24170; Abcam), phospho-ULK1 (S757) (6888; Cell Signaling Technology), phospho-S6 kinase (T389) (9202; Cell Signaling Technology), and tubulin (3873; Cell Signaling Technology).

### Cell culture and induction of autophagy

Newt cells were seeded on 12-mm-diameter cover glasses in a 6-cm dish with the culture medium described above, cultured for 24 h, washed twice with PBS, and subjected to starvation using starvation medium specialized for newt for the indicated duration. To make 100 ml of starvation medium, 65 ml of Earle's Balanced Salt Solution (E2888; Sigma-Aldrich) and 35 ml of distilled water were mixed. The ideal working concentration of bafilomycin A1 for both mammalian and newt cells was verified through appropriate LC3 accumulation in an autophagy flux assay.

### Immunofluorescence staining and microscopic observation

For immunostaining, newt cells were cultured on fibronectin (2.5 $\mu g/cm^2$)-coated 12-mm-diameter cover glasses in a 6-cm dish and fixed with 4% PFA. After permeabilization with 50 $\mu g/ml$ of digitonin in PBS for 10 min, the cells were blocked with PBS supplemented with 0.2% gelatin for 20 min, incubated with primary antibodies, washed twice with PBS, and incubated with secondary antibodies and DAPI in PBS supplemented with 0.2% gelatin for 60 min. Coverslips were mounted with ProLong Gold antifade reagent (P36934; Invitrogen) and observed using a fluorescence microscope (IX71 and DP80; Olympus). The obtained images were analyzed for the number of autophagosomes, lysosomes, nuclei, and proteins via mTOR staining using ImageJ imaging software. To visualize lysosomes, cells were cultured in the presence of 50 nM LysoTracker Red DND-99 (L7528; Thermo Fisher Scientific) for 30 min before harvesting.

### Western blotting

Newt cell lysates were obtained with SDS sample buffer, loaded on SDS–PAGE gels, and transferred to PVDF membranes. The membranes were stained with Ponceau S, washed with distilled water and TBST, blocked with 1% skim milk in TBST for 30 min, and incubated with primary antibodies in TBST supplemented with 1% skim milk for 1 h at room temperature. After washing three times with TBST, the membranes were incubated with secondary antibodies conjugated with HRP. Western chemiluminescent signals were obtained using ImmunoStar Zeta or ImmunoStar LD (295-72404 and 290-69904, respectively; Fujifilm Wako Pure Chemical) and detected with ImageQuant LAS 4000 (Cytiva).

### Statistical analysis

Quantitative data are presented as the mean ± SD unless otherwise stated. The statistical analyses were performed using ordinary one-way ANOVA for multiple comparisons.

# Data Availability

The datasets used and/or analyzed during the current study are available from the corresponding author upon reasonable request.

# Supplementary Information

# Acknowledgements

We thank Li laboratory members for the discussion and technical support for the maintenance of newts. This study was supported by grants from JSPS KAKENHI (19H03454, 23K18219).

## Author Contributions

MM Hasan: resources, investigation, methodology, and writing—original draft, review, and editing.
S Goto: resources and methodology.
R Sekiya: resources and methodology.
T Hayashi: resources and methodology.
T-S Li: conceptualization, resources, supervision, and methodology.
T Kawabata: conceptualization, data curation, formal analysis, supervision, funding acquisition, validation, investigation, visualization, methodology, project administration, and writing—original draft, review, and editing.

## Conflict of Interest Statement

The authors declare that they have no conflict of interest.

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
