## [Reviewer comments · Life Science Alliance]

Life Science Alliance

Sustained induction of autophagy enhances survival during prolonged starvation in newt cells

Md. Hasan, Shinji Goto, Reiko Sekiya, Toshinori Hayashi, Tao-Sheng Li, and Tsuyoshi Kawabata

DOI: <https://doi.org/10.26508/lsa.202402772>

Corresponding author(s): *Tsuyoshi Kawabata, Nagasaki University*

Review Timeline:

Submission Date:	2024-04-17
Editorial Decision:	2024-06-17
Revision Received:	2024-12-29
Editorial Decision:	2025-01-21
Revision Received:	2025-01-28
Accepted:	2025-01-29

Transaction Report:

June 17, 2024

Re: Life Science Alliance manuscript #LSA-2024-02772

Dr. Tsuyoshi Kawabata
Nagasaki University
Department of Stem Cell Biology
1-12-4 Sakamoto
Nagasaki, Nagasaki 852-8523
Japan

Dear Dr. Kawabata,

Thank you for submitting your manuscript entitled "Constitutive activation of autophagy promotes survival during prolonged starvation in newt cells" to Life Science Alliance. The manuscript was assessed by expert reviewers, whose comments are appended to this letter. We invite you to submit a revised manuscript addressing the Reviewer comments.

Thank you for this interesting contribution to Life Science Alliance. We are looking forward to receiving your revised manuscript.

Sincerely,

B. MANUSCRIPT ORGANIZATION AND FORMATTING:

Reviewer #1 (Comments to the Authors (Required)):

GENERAL REMARKS:

The study discusses the role of autophagy in promoting the remarkable starvation resistance observed in salamanders, using the newt *Pleurodeles waltl* as a model organism. Understanding the mechanisms by which salamanders, and potentially other organisms with similar adaptations, withstand prolonged periods of starvation could have significant implications for our understanding of survival strategies in diverse biological systems. Thus, it is a very good attempt at working on something novel that could have therapeutic implications in human health and disease.

The manuscript is easy to read and understand, the objectives of the study and the novelty of the work is put forth well. That being said, the results of the studies do not seem to be conclusive enough to put forward the claims that this study is making.

Specific points:

1. There are some typing errors that are overlooked. For eg: Newt cells in line 134 and 421. The authors can reread the draft and correct the typing errors.

2. Figure 1: B. and C. One Way Anova is used, but which test is used for the comparison is not mentioned.

Since the size of the newt cells are bigger, should the number of autophagosomes be also normalised to cell size, for both mammalian cells and the newt cells to have an even comparison?

3. Figure 2B

(i) Shouldn't labelling the Y-axis as 'number of autophagosomes' better than labelling 'number of LC3 dots'?

(ii) 4h seems like a more important time point from Fig 1, so, Figure 2 should have had 4h time point as well, in addition to the 2h and 24h

(iii) SEM of three independent images is not enough to quantify (assuming they don't mean three independent experiments). Number of cells used for quantification should be specified.

(iv) Moreover, the data should be summarized with Standard Deviation (SD) instead of Standard Error of Mean (SEM) (Applicable to all statistics done in the manuscript)

4. Fig S1 A. and B.: The LC3-II expression seems to go down in EBSS condition (without Baf treatment) between no treatment and 2h in EBSS. Shouldn't the levels of LC3-II go up at 2h?

5. Fig 3A: Scale bars missing in images.

6. Fig 3: As opposed to using LysoTracker and GFP-LC3, tandem GFP-RFP-LC3 construct could be used to bring the point across in a better way.

7. Fig 3B: Y-axis on the bar graph could be labelled as colocalization instead of/in addition to Mander's coefficient. 3 images don't suffice to do statistical analysis on microscopy data. Number of cells should be mentioned.

8. Fig 4B: The bar graph does not seem to match the representative images in the sense that the colocalization gradually seem to go down in the pictures in the starvation condition.

Moreover, the decrease in colocalization could be because of the decline in signal intensity of the lysosomes, as shown by using

lysotracker. So, maybe the colocalization could be normalized to number of lysosomes, and percent colocalization be plotted.

9. Fig S2 B: The Y-axis of the bar graph is 'relative size of lysosomes', but what is it relative to, is not clear.

Reviewer #2 (Comments to the Authors (Required)):

The manuscript by Hassan MM et al., investigates the autophagy machinery in newt cells in comparison to mammalian cells to determine the mechanism that contributes to starvation hyper-resistance in newts. Although the initial findings of the paper seem interesting, the data are too preliminary to support the author's claim that the newt cells maintain constitutive activation of autophagy during prolonged starvation due to its unique regulation of mTOR. The data needs to be extensively validated with additional experiments. These limitations make the manuscript unsuitable for publication in Life Science Alliance.

Comments:

1. The primary conclusion "constitutive activation of autophagy promotes survival" is not supported by the evidence provided. The western blot in Figure S1B as well as Figure 1, shows that autophagosomes decrease over time in newt cells. In Figure S3, the authors report the relative survival of newt cells during starvation. It is important to provide similar data in mammalian cells. Furthermore, to determine the importance of autophagy for survival, the authors should also include the treatment of genetic or chemical intervention to autophagy and compare the survival between newt and mammalian cells.
2. Since the whole manuscript is a cross-comparison study between newt and mammalian cells, it is essential to provide suitable as well as consistent evidence in the mammalian cells. All the results provided for newt cells must be shown with the direct corresponding comparisons in mammalian cells. For instance, the timepoints in figure are not same as in figure 1B, especially since the 4h is the peak timepoint in newt cells. Figure 1C, Figure 3 and 4 are missing for mammals. This makes most of the results of the manuscript as a statement rather than an evidence based conclusion.
3. The measurement of autophagy kinetics in cell lines that are cultivated under different temperature conditions must be carefully evaluated as rightly pointed out by the authors themselves. The authors should also perform the starvation experiments under the maximum and minimum temperature at which newt and mammalian cells can be cultivated, and observe or rule-out the effect of temperature on autophagy kinetics.
4. Figure 2B, error bar: SEM of at least 3 independent images. Is this the same case for Figure 1? This is very little data to make any observation and is quite unconventional- observations from at least 3 independent experiments are needed. Each experiment should contain 3 technical replicates and each replicate should count many cells.
5. "Line 127-128: autophagy is slower and are retained longer than that in normal mammalian cells; Line: 193-194: "We have shown that the biogenesis of autophagosomes is higher in newt cells than in mammalian cells during a prolonged period of starvation". Throughout the manuscript, it is unclear whether the induction of autophagosomes are higher and prolonged in newt cells, or the clearance of autophagosomes are slower in newt cells in comparison to mammalian cells. The authors should clarify this point with additional experiments, such as using more markers and interventions to determine efficiency of various stages of autophagy, such as initiation, elongation and substrate-selection, lysosome maturation, and fusion of newt cells in comparison to mammalian cells.
6. Line: 199-205: "We observed that the acute induction rate of autophagy in starved newt cells, as measured by Western blotting, was comparable to that in mammalian cells (Fig. 1A and B)". First of all, the timepoints of comparison are not exactly the same and the scale is different. Furthermore, the western blot needs to be quantified, otherwise it is hard to observe the changes. Looking at the LC3 quantification in newt cells, until 2 h there is no significant induction of autophagy, but it increases rapidly afterward, peaking around 4 h. Whereas in mammals, there is already a significant induction of autophagy at 2 h. This suggests that the autophagy induction kinetics are also different between the two. Hence the statements such as in Line 126-127: "These results indicate that newt cells induce autophagy in response to starvation, as in mammalian cells"; Line 201-205: "Furthermore, the induction of autophagosome biogenesis by the starvation signal reached its highest level in newt cells 4 h after induction, which is comparable to that in mammalian cells (Fig. 204 1A and B), suggesting that the higher autophagic activity of newt cells exposed to prolonged starvation is not mainly due to a delayed response" are not entirely correct.
7. Line 137-139: "We found that a significant proportion of newt cells were larger than 4 μm , suggesting that newts may have a relatively higher efficiency of autophagic activity than mammalian cells". The authors report that the autophagosomes in the newt cells are larger than in the mammalian cells and suggest that newts may have a relative higher efficiency of autophagic activity than mammalian cells. However, the newts cells are also larger in comparison to mammalian cells, as pointed out by the authors. So, the authors should consider quantifying the size of the autophagosomes relative to the cell size and use additional measures to determine the efficiency of autophagic activity.
8. "Line 159-161: we found that mTOR colocalised with the LysoTracker signal under nutrient-rich conditions and dissociated

from the lysosome 30 min after induction of starvation (Figure 4A and B)". This is an interesting observation; however, the authors did not include similar data for mammalian cells. Notably, this observation in newt cells is different from the previous observation in axolotl mTOR, which is reported to be less-sensitive to starvation conditions (Zhulyn O et al., 2023, using a modified human mTOR). However, the authors did not elaborate further on this finding and the differences between the two studies, whether it is due to the chimeric construct or the micro-environment between the mammalian and salamander cell lines, etc. Due to the differences in these observations and the lack of a similar comparison in mammalian cells, the authors should include additional experiments, such as similar mTOR - lysosome localization during starvation in mammalian cells, prolonging the starvation to more than 24 h to determine the time-point at which the mTOR activity gets restored in newt cells; including observation of additional substrate of mTOR such as 4E-BP1 phosphorylation, observation of autophagy kinetics upon inhibition of mTOR (using rapamycin) under nutrient-rich conditions, amino-acid titration experiments, and mTOR-lysosome localization kinetics to determine the differences in the response between newt and mammalian cells

9. The authors did not discuss whether basal activity of autophagy and the mTOR are different or similar under nutrient-rich conditions between newt and mammalian cells

Reviewer #1 (Comments to the Authors (Required)):

GENERAL REMARKS:

The study discusses the role of autophagy in promoting the remarkable starvation resistance observed in salamanders, using the newt *Pleurodeles waltl* as a model organism. Understanding the mechanisms by which salamanders, and potentially other organisms with similar adaptations, withstand prolonged periods of starvation could have significant implications for our understanding of survival strategies in diverse biological systems. Thus, it is a very good attempt at working on something novel that could have therapeutic implications in human health and disease.

The manuscript is easy to read and understand, the objectives of the study and the novelty of the work is put forth well. That being said, the results of the studies do not seem to be conclusive enough to put forward the claims that this study is making.

Specific points:

1. There are some typing errors that are overlooked. For eg: Newt cells in line 134 and 421. The authors can reread the draft and correct the typing errors.

>Thank you for the comment. We corrected the errors.

2. Figure 1: B. and C. One Way Anova is used, but which test is used for the comparison is not mentioned.

>We added detailed information about the tests.

Since the size of the newt cells are bigger, should the number of autophagosomes be also normalised to cell size, for both mammalian cells and the newt cells to have an even comparison?

>The reviewer correctly noted that newt cells are considerably larger than the mammalian cell lines utilized in this research. Given that cell size varies even among different mammalian cell lines, comparing the number of autophagosomes per cell across different cell lines lacks significant biological relevance. To accurately assess the autophagy-inducing capacity of newt cells under starvation conditions, we presented the autophagosome count relative to cell area, enabling a more appropriate comparison among the various cell types examined in this study.

3. Figure 2B

(i) Shouldn't labelling the Y-axis as 'number of autophagosomes' better than labelling 'number of LC3 dots'?

>We agree with this idea. the labeling has been changed.

(ii) 4h seems like a more important time point from Fig 1, so, Figure 2 should have had 4h time point as well, in

addition to the 2h and 24h

>Thank you for the comment. We agree with this idea that the 4h time point is important for the comparison. We carried out experiments for this matter and added new data in the revised manuscript.

(iii) SEM of three independent images is not enough to quantify (assuming they don't mean three independent experiments). Number of cells used for quantification should be specified.

>We apologize for the error. We performed three independent experiments to obtain these data. At least 30 independent images, with a total of at least 200 cells, were analyzed for each experimental group.

(iv) Moreover, the data should be summarized with Standard Deviation (SD) instead of Standard Error of Mean (SEM) (Applicable to all statistics done in the manuscript)

>We appreciate the comment. To visualize actual experimental variations, we used box plots which show more accurate variations and relevant to SD in the revised manuscript.

4. Fig S1 A. and B.: The LC3-II expression seems to go down in EBSS condition (without Baf treatment) between no treatment and 2h in EBSS. Shouldn't the levels of LC3-II go up at 2h?

>We acknowledge that the concept can be perplexing, but a growing consensus suggests that an increase in LC3-II doesn't necessarily indicate heightened autophagic flux. This is because the lipidated form of LC3 is recruited both inside and outside the autophagosome, subjecting it to degradation through autophagy itself. Consequently, LC3-II levels may decrease (or increase if LC3 synthesis and lipidation activity exceed the rise in autophagic LC3 degradation) in cells experiencing high autophagic activity induced by starvation. To accurately assess autophagic activity, an autophagic flux assay utilizing lysosomal inhibitors is essential. We have incorporated an explanation of this phenomenon in the manuscript.

5. Fig 3A: Scale bars missing in images.

>Thank you for the comment. We added scale bars in all the images in this study.

6. Fig 3: As opposed to using LysoTracker and GFP-LC3, tandem GFP-RFP-LC3 construct could be used to bring the point across in a better way.

>We agree that tandem-tagged LC3 is a better way to estimate autophagic activity in these cells. However, it is difficult to use this construct because of the technical difficulty of newt primary cells. Indeed, mammalian proteins cannot be expressed in newt cells, possibly because of differences in codon usage between them. Thus, we tried new reagents supplied by DOJINDO (Autophagic Flux Assay Kit, 348-10101) which can distinguish between autophagosomes and autolysosomes. Using this reagent, we obtained new data showing that newt cells retain proper autophagic activity even 24 h after starvation (Figure S7).

7. Fig 3B: Y-axis on the bar graph could be labelled as colocalization instead of/in addition to Mander's coefficient.

>Thank you for the comment. We changed the word.

3 images don't suffice to do statistical analysis on microscopy data. Number of cells should be mentioned.

>We apologize for our error in this matter as well. For each dataset, a minimum of 20 cells were examined. We have rectified the typographical error and included new experiments with a 4-hour time point.

8. Fig 4B: The bar graph does not seem to match the representative images in the sense that the colocalization gradually seem to go down in the pictures in the starvation condition.

Moreover, the decrease in colocalization could be because of the decline in signal intensity of the lysosomes, as shown by using lysotracker. So, maybe the colocalization could be normalized to number of lysosomes, and percent colocalization be plotted.

>The experiment was conducted again, with a subsequent reanalysis of the data and an update to the representative images (now corresponding to Figure 3B in the revised manuscript). We quantified lysosome numbers and incorporated a summary box plot (Figure 2D), which demonstrates a significant increase in lysosome count under starvation conditions compared to nutrient-rich environments. In the updated data presented in the revised manuscript, no distinct differences in the colocalization index were observed across the NT, 2, 4, and 24-hour timepoints. We hypothesize that the increased lysosome count in starved newt cells may contribute to enhanced colocalization between mTOR and lysosomes. This observation has been noted in the manuscript as follows.

"Conversely, newt cells exhibit only partial PlemTOR localization on lysosomes, even in nutrient-rich conditions (Figure 3B and C). Moreover, PlemTOR's lysosomal localization did not significantly change during starvation for up to 24 hours. This could be partly attributed to increase in lysosome number, but not size, during starvation in newt cells (Figure 2D and Figure S8). Consequently, PlemTOR's lysosomal localization may be underestimated in nutrient-rich conditions. Nevertheless, the lack of changes in PlemTOR's lysosomal localization during starvation might indicate that mTOR is starvation-resistant, as suggested by a previous study showing that changes in lysosomal localization of Axolotl-mimic mTOR are less sensitive than mammalian mTOR when expressed in mammalian cells(Zhulyn et al, 2023). "

9. Fig S2 B: The Y-axis of the bar graph is 'relative size of lysosomes', but what is it relative to, is not clear.

>Thank you for the comment. It is relative to NT condition. I updated the information.

Reviewer #2 (Comments to the Authors (Required)):

The manuscript by Hassan MM et al., investigates the autophagy machinery in newt cells in comparison to mammalian cells to determine the mechanism that contributes to starvation hyper-resistance in newts. Although the initial findings of the paper seem interesting, the data are too preliminary to support the author's claim that the newt cells maintain constitutive activation of autophagy during prolonged starvation due to its unique regulation of mTOR. The data needs to be extensively validated with additional experiments. These limitations make the manuscript unsuitable for publication in Life Science Alliance.

Comments:

1. The primary conclusion "constitutive activation of autophagy promotes survival" is not supported by the evidence provided. The western blot in Figure S1B as well as Figure 1, shows that autophagosomes decrease over time in newt cells. In Figure S3, the authors report the relative survival of newt cells during starvation. It is important to provide similar data in mammalian cells. Furthermore, to determine the importance of autophagy for survival, the authors should also include the treatment of genetic or chemical intervention to autophagy and compare the survival between newt and mammalian cells.

>Thank you for the comment. We have added new mammalian data for comparison between newts and mammals for autophagosome formation, mTOR localization, mTOR substrate phosphorylation, cell survival, and colocalization of LC3 with lysosomes. In addition, we performed a survival assay using the lysosomal inhibitor bafilomycin A1 to determine the role of autophagy in survival during starvation in both newts and mammalian cells.

2. Since the whole manuscript is a cross-comparison study between newt and mammalian cells, it is essential to provide suitable as well as consistent evidence in the mammalian cells. All the results provided for newt cells must be shown with the direct corresponding comparisons in mammalian cells. For instance, the timepoints in figure are not same as in figure 1B, especially since the 4h is the peak timepoint in newt cells. Figure 1C, Figure 3 and 4 are missing for mammals. This makes most of the results of the manuscript as a statement rather than an evidence based conclusion.

>We agree with the reviewer's suggestion. In the revised manuscript, we have added a line of datasets obtained using two mammalian cell lines, human dermal fibroblasts and PANC1 (pancreas cancer cell line) to compare newts and mammals.

3. The measurement of autophagy kinetics in cell lines that are cultivated under different temperature conditions must be carefully evaluated as rightly pointed out by the authors themselves. The authors should also perform the starvation experiments under the maximum and minimum temperature at which newt and mammalian cells can be cultivated, and observe or rule-out the effect of temperature on autophagy kinetics.

>Thank you for your comments. We consider that the difference in adequate temperature between newts and mammalian cells is critical. As the increase in temperature for newt cell culture is highly cytotoxic, we attempted to conduct autophagic kinetics analysis using mammalian cells under lower temperatures than their adequate condition; although it is known that low temperature causes cold stress, we expected the stress might change the proper autophagic activity. At 25 °C, mammalian HDF showed an increase in the number of autophagosomes, even under nutrient-rich conditions, than at 37 °C. The number of autophagosomes/area in nutrient-rich conditions at 25 °C reached almost the same level as the highest number in starved HDF at 37 °C, indicating that cold stress masked the induction of autophagosome formation by starvation in HDFs. This was also similar in the case of PANC1 cells, but did not appear to be more severely affected than in HDFs. Overall, we could not clearly show the effect of temperature on autophagosome formation in mammalian cells because each organism is optimized to its living environment. However, we could at least mention that there was no clear change in the rapid induction of autophagosome formation at 2 h after starvation, but newt cells exhibited higher autophagosome formation activity

at 96 h post-starvation. We have updated the text and the results. We appreciate your comment and believe that the revised datasets and discussions have been greatly improved to understand the difference between newt and mammalian cells.

4. Figure 2B, error bar: SEM of at least 3 independent images. Is this the same case for Figure 1? This is very little data to make any observation and is quite unconventional- observations from at least 3 independent experiments are needed. Each experiment should contain 3 technical replicates and each replicate should count many cells.

>We apologize for the error. We performed three independent experiments to obtain these data. At least 30 independent images, with a total of at least 200 cells, were analyzed for each experimental group.

5. "Line 127-128: autophagy is slower and are retained longer than that in normal mammalian cells; Line: 193-194: "We have shown that the biogenesis of autophagosomes is higher in newt cells than in mammalian cells during a prolonged period of starvation". Throughout the manuscript, it is unclear whether the induction of autophagosomes are higher and prolonged in newt cells, or the clearance of autophagosomes are slower in newt cells in comparison to mammalian cells. The authors should clarify this point with additional experiments, such as using more markers and interventions to determine efficiency of various stages of autophagy, such as initiation, elongation and substrate-selection, lysosome maturation, and fusion of newt cells in comparison to mammalian cells.

>We appreciate this comment and agree that the kinetics of autophagosome maturation and autolysosome formation should be tested to understand the kinetics of autophagy in newt cells. However, owing to limitations in the genetic approach of newt cells and the low reactivity of commercially available antibodies used for autophagy analysis, it is difficult to conduct conventional methods for newt cells. To this end, we utilized new reagents supplied by DOJINDO (Autophagic Flux Assay Kit, 348-10101) which can distinguish between autophagosomes and autolysosomes. Using this reagent, we obtained new data showing that newt cells retained proper autophagic activity even 24 h after starvation (Figure S7). This suggests that autolysosome maturation is upregulated in newt cells at 4, 8, and 24 h after starvation induction. Moreover, the kinetics of ULK1 dephosphorylation during starvation supports this hypothesis (Figure 3D), as this event is the upstream regulation of autophagy. Therefore, we consider that the number of LC3 dots under starved conditions correctly reflects autophagic activity.

6. Line: 199-205: "We observed that the acute induction rate of autophagy in starved newt cells, as measured by Western blotting, was comparable to that in mammalian cells (Fig. 1A and B)". First of all, the timepoints of comparison are not exactly the same and the scale is different.

>We performed LC3 dot formation assay for both newt and mammalian cells with the same time points. As the size of newt and mammalian cells are different, we showed the data as the number of autophagosome per cellular area. The direct comparison of the size of newt and HDFs were also shown in Figure S4.

Furthermore, the western blot needs to be quantified, otherwise it is hard to observe the changes. Looking at the LC3 quantification in newt cells, until 2 h there is no significant induction of autophagy, but it increases rapidly afterward, peaking around 4 h. Whereas in mammals, there is already a significant induction of autophagy at 2 h. This suggests that the autophagy induction kinetics are also different between the two. Hence the statements such as in Line 126-127: These results indicate that newt cells induce autophagy in response to starvation, as in mammalian

cells"; Line 201-205: "Furthermore, the induction of autophagosome biogenesis by the starvation signal reached its highest level in newt cells 4 h after induction, which is comparable to that in mammalian cells (Fig. 204 1A and B), suggesting that the higher autophagic activity of newt cells exposed to prolonged starvation is not mainly due to a delayed response" are not entirely correct.

>We apologize for this confusing statement in the manuscript. In the revised manuscript, depending on additional data, we changed description about the difference in autophagic induction between newt cells and mammalian cells.

7. Line 137-139: "We found that a significant proportion of newt cells were larger than 4 μm , suggesting that newts may have a relatively higher efficiency of autophagic activity than mammalian cells". The authors report that the autophagosomes in the newt cells are larger than in the mammalian cells and suggest that newts may have a relative higher efficiency of autophagic activity than mammalian cells. However, the newts cells are also larger in comparison to mammalian cells, as pointed out by the authors. So, the authors should consider quantifying the size of the autophagosomes relative to the cell size and use additional measures to determine the efficiency of autophagic activity.

>We acknowledge this feedback and concur with the notion that comparing autophagosome size and autophagic activity between newt and mammalian cells is challenging due to their size differences. Given the significant variation in cell sizes even among mammalian cells (e.g., lymphocytes at $130\mu\text{m}^3$, fibroblasts at $2000\mu\text{m}^3$, and cardiomyocytes at $15000\mu\text{m}^3$), it may not be appropriate to assess autophagic activity by simply normalizing autophagosome size to cell size. To avoid confusion, we have revised our manuscript to state that autophagosomes in newt cells are larger than those in the mammalian cells examined in this study. This suggests that each autophagosome in newt cells might have a greater capacity to engulf cytoplasmic components compared to those in mammalian cells.

8. "Line159-161: we found that mTOR colocalised with the LysoTracker signal under nutrient-rich conditions and dissociated from the lysosome 30 min after induction of starvation (Figure 4A and B)". This is an interesting observation; however, the authors did not include similar data for mammalian cells. Notably, this observation in newt cells is different from the previous observation in axolotl mTOR, which is reported to be less-sensitive to starvation conditions (Zhulyn O et al., 2023, using a modified human mTOR). However, the authors did not elaborate further on this finding and the differences between the two studies, whether it is due to the chimeric construct or the micro-environment between the mammalian and salamander cell lines, etc. Due to the differences in these observations and the lack of a similar comparison in mammalian cells, the authors should include additional experiments, such as similar mTOR - lysosome localization during starvation in mammalian cells, prolonging the starvation to more than 24 h to determine the time-point at which the mTOR activity gets restored in newt cells; including observation of additional substrate of mTOR such as 4E-BP1 phosphorylation, observation of autophagy kinetics upon inhibition of mTOR (using rapamycin) under nutrient-rich conditions, amino-acid titration experiments, and mTOR-lysosome localization kinetics to determine the differences in the response between newt and mammalian cells

>This comment is highly valued. We acknowledge the previous report indicating that Axolotl mTOR exhibits a less sensitive response to starvation compared to mammalian mTOR. Given the close relationship between newts and Axolotls, and the high similarity in their mTOR amino acid sequences, we anticipate that newts might also display a

reduced sensitivity to starvation in terms of mTOR-dependent regulation of downstream pathways, including autophagy. Our additional data revealed that mTOR localization on lysosomes changed only slightly in response to starvation. We believe this doesn't necessarily indicate mTOR inactivity in newt cells and may be partially attributed to differences in mTOR regulation, as only a portion of mTOR localizes on lysosomes in newts even under nutrient-rich conditions. This contrasts with mammalian cells, which show almost complete mTOR localization on lysosomes in nutrient-rich conditions (Fig. 3B and C). Notably, the phosphorylation of ULK1, an autophagy regulator negatively controlled by mTOR through phosphorylation, is dephosphorylated under starvation conditions, comparable to mammalian cells (Fig. 3B and C). Interestingly, as shown in the revised manuscript, newts possess an alternative mTOR ortholog variant that includes insertion 2 but lacks insertion 1, which is the key factor determining Axolotl mTOR activity. We propose that newt and Axolotl mTOR may share regulatory features distinct from mammals, while newts may also maintain unique characteristics different from Axolotls. This could explain newts' ability to achieve both starvation resistance and high tissue regenerative capacity. We have incorporated this discussion into the revised manuscript.

9. The authors did not discuss whether basal activity of autophagy and the mTOR are different or similar under nutrient-rich conditions between newt and mammalian cells

>We appreciate your feedback and concur with the notion of a disparity in basal mTOR activity between newts and mammals. The Axolotl study you referenced explored this concept by examining mTOR expression levels in tissue samples using antibody against mTOR. This topic presents challenges due to variations in antibody reactivity (the study utilized antibodies generated against human mTOR epitopes) and differences in tissue structure. Nevertheless, we have incorporated a discussion in the revised manuscript addressing the disparities in baseline mTOR and autophagy activity between newts and mammals.

January 21, 2025

RE: Life Science Alliance Manuscript #LSA-2024-02772R

Dr. Tsuyoshi Kawabata
Nagasaki University
Department of Stem Cell Biology
1-12-4 Sakamoto
Nagasaki, Nagasaki 852-8523
Japan

Dear Dr. Kawabata,

Thank you for submitting your revised manuscript entitled "Constitutive activation of autophagy promotes survival during prolonged starvation in newt cells". We would be happy to publish your paper in Life Science Alliance pending final revisions necessary to meet our formatting guidelines.

- please address Reviewer 2's numbered points
- please be sure that the authorship listing and order is correct
- please upload all figure files as individual ones, including the supplementary figure files; all figure legends should only appear in the main manuscript file
- please remove figures from the manuscript file and leave them uploaded separately
- upload a clean manuscript file without highlighted changes
- please add the Twitter handle of your host institute/organization as well as your own or/and one of the authors in our system
- please be sure that the authorship listing and order is correct
- please move your main and supplementary figure legends to the main manuscript text after the references section
- please add an Author Contributions section to your main manuscript text
- please add a Conflict of Interest statement to your main manuscript text
- please add callouts for Figures 2C; 4D-F; S5A,B; S7A,B; S8A,B to your main manuscript text

FIGURE CHECK:

- please provide the size of the scale bars in Figure S5A
- please add sizes next to all blots
- there does not appear to be a Figure S6. please update the numbering and labeling of the supplemental figures, including the callouts within the text.

A. FINAL FILES:

B. MANUSCRIPT ORGANIZATION AND FORMATTING:

Sincerely,

Reviewer #2 (Comments to the Authors (Required)):

The manuscript by Hassan MM et al. has been significantly improved from the previous version and conveys the following key observations:

1. Both mammalian and newt cells rapidly induce autophagy in response to nutrient starvation. However, newt cells display slightly slower and more gradual induction kinetics of autophagy compared to mammalian cells.
 2. Newt mTOR exhibits reduced localization to lysosomes but still regulates downstream proteins in a manner similar to mammalian cells, highlighting its unique regulatory mechanisms.
- This paper is of great interest to a broader audience and emphasizes the importance of including and utilizing non-traditional model organisms, such as salamander species, to explore their unique adaptations and the fundamental mechanisms of biology.

That being said, even though the mTOR part has been improved by the authors, it still does not provide mechanistic insights into the regulation. However, if this is still aligns with the scope of the journal, the article can be accepted after the authors address the following points:

1. Please review the document thoroughly and correct any spelling mistakes.
2. Add figure references next to their respective descriptions in the text for easier readability. Also please check the citations, whether they are included next to the corresponding text.
3. Include the number of experiments conducted (e.g., "n") in the figure legends. For example, specify this for Figures S1A and S1B.
4. Figure S4 Analysis: Since newt cells already exhibit a larger initial autophagosome size, the cross-species comparison in panel B is less informative. Instead, consider showing how autophagosome size changes during starvation in other species,

similar to what is done in panel A for newt cells.

5. Line 207 Clarification: The manuscript states that baseline mTOR activity in axolotls is very low, not that mTOR expression is higher in axolotls compared to mammals. It also specifies that axolotl mTOR exists in a "primed" hypersensitive state, allowing rapid activation. Please clarify this point for accuracy.

6. Since this study involves a cross-species comparison, provide details about the culture media used for each cell line. For example, how the differences in glucose concentration may influence the observed outcomes. Discuss these differences in the manuscript or mention them as a limitation of the study.

7. BafA1 Concentration: How was the concentration of bafilomycin A1 determined? Since you are comparing two different cell types (newt cells vs. mammalian cells), was the concentration optimized for both? Please mention in the methodology.

8. Lysosome Biogenesis in Mammalian Cells: Since the increase in lysosome numbers appears to play a significant role in the observations for newt cells, was a similar increase observed in mammalian cells during starvation?

Reviewer #2 (Comments to the Authors (Required)):

The manuscript by Hassan MM et al. has been significantly improved from the previous version and conveys the following key observations:

1. Both mammalian and newt cells rapidly induce autophagy in response to nutrient starvation. However, newt cells display slightly slower and more gradual induction kinetics of autophagy compared to mammalian cells.
2. Newt mTOR exhibits reduced localization to lysosomes but still regulates downstream proteins in a manner similar to mammalian cells, highlighting its unique regulatory mechanisms.

This paper is of great interest to a broader audience and emphasizes the importance of including and utilizing non-traditional model organisms, such as salamander species, to explore their unique adaptations and the fundamental mechanisms of biology.

That being said, even though the mTOR part has been improved by the authors, it still does not provide mechanistic insights into the regulation. However, if this still aligns with the scope of the journal, the article can be accepted after the authors address the following points:

>We are deeply grateful for your exceptional effort in reviewing our manuscript. Your valuable input has significantly enhanced the quality of our work.

1. Please review the document thoroughly and correct any spelling mistakes.

>We found some typos and corrected them in the revised manuscript.

2. Add figure references next to their respective descriptions in the text for easier readability. Also please check the citations, whether they are included next to the corresponding text.

>Thank you for the suggestion. We changed some figure references and citations to improve the readability.

3. Include the number of experiments conducted (e.g., "n") in the figure legends. For example, specify this for Figures S1A and S1B.

>We added the number of experiments in the figure legends.

4. Figure S4 Analysis: Since newt cells already exhibit a larger initial autophagosome size, the cross-species comparison in panel B is less informative. Instead, consider showing how autophagosome size changes during starvation in other species, similar to what is done in panel A for newt cells.

>We added the data showing the changes in the size of autophagosome in mammalian cells during

starvation (Figure S4).

5. Line 207 Clarification: The manuscript states that baseline mTOR activity in axolotls is very low, not that mTOR expression is higher in axolotls compared to mammals. It also specifies that axolotl mTOR exists in a "primed" hypersensitive state, allowing rapid activation. Please clarify this point for accuracy.

>Thank you for the suggestion. We changed this sentence in the revised manuscript.

6. Since this study involves a cross-species comparison, provide details about the culture media used for each cell line. For example, how the differences in glucose concentration may influence the observed outcomes. Discuss these differences in the manuscript or mention them as a limitation of the study.

>Newt cells should be cultured in a different osmotic pressure from mammalian cells. Some nutrient factors such as insulin may alter the phenotype. We added explanation in the discussion section.

7. BafA1 Concentration: How was the concentration of bafilomycin A1 determined? Since you are comparing two different cell types (newt cells vs. mammalian cells), was the concentration optimized for both? Please mention in the methodology.

>The effectiveness of Bafilomycin A1 was verified through an LC3 flux assay, which demonstrated LC3 accumulation due to lysosomal inhibition. We have included additional details about this process in the materials and methods section.

8. Lysosome Biogenesis in Mammalian Cells: Since the increase in lysosome numbers appears to play a significant role in the observations for newt cells, was a similar increase observed in mammalian cells during starvation?

>We added information about the changes in the number of lysosomes in mammalian cells which showed only subtle change during starvation (Figure S7).

January 29, 2025

RE: Life Science Alliance Manuscript #LSA-2024-02772RR

Dr. Tsuyoshi Kawabata
Nagasaki University
Department of Stem Cell Biology
1-12-4 Sakamoto
Nagasaki, Nagasaki 852-8523
Japan

Dear Dr. Kawabata,

Thank you for submitting your Research Article entitled "Sustained induction of autophagy enhances survival during prolonged starvation in newt cells". It is a pleasure to let you know that your manuscript is now accepted for publication in Life Science Alliance. Congratulations on this interesting work.

DISTRIBUTION OF MATERIALS:

Again, congratulations on a very nice paper. I hope you found the review process to be constructive and are pleased with how the manuscript was handled editorially. We look forward to future exciting submissions from your lab.

Sincerely,
